# God's Moral Perfection as His Beneficent Love. Comment on Craig (2023). Is God's Moral Perfection Reducible to His Love? *Religions* 14: 140

**Kevin Kinghorn** 

Asbury Theological Seminary, Wilmore, KY 40390, USA; kevin.kinghorn@asburyseminary.edu

**Abstract:** William Lane Craig insists that I am wrong in reducing God's moral goodness to his beneficent aim of drawing all people to himself. For Craig, God's moral goodness, best conceived in terms of righteousness, must also include God's retributive justice toward the wicked, who deserve the punishment they receive. My response is that Craig's argument rests on two assumptions about value, neither of which, I argue, Christian theists have good reason to affirm.

**Keywords:** retributive justice; desert; divine goodness; righteousness; divine love; final value



## 1. Introduction

In a recent article (Craig 2023), William Lane Craig criticizes my claim that all God's actions toward us are motivated by God's beneficent love for us. Craig focuses on my book, *But What About God's Wrath?* (Kinghorn 2019), in which I looked at various biblical passages that describe God as acting in wrath toward people. I argued that the biblical depictions of God's wrath are best understood as God "pressing upon people difficult truths about themselves". These divine acts can be extremely painful to us, and they are acts of last resort. But they remain one means God has for prompting repentance so that individuals can be reconciled to God and to one another. Thus, the actions of God described in the Bible as wrathful would be intended as ultimately restorative, consistent with the beneficent concern that, I claim, motivates all God's actions toward us.

Craig objects that such a single motivation cannot adequately account for the biblical picture of God's *righteousness*. In response to immorality, this righteousness manifests itself in punishment of the wicked. It is the wicked who have opposed God and have oppressed God's people. And God's vindication of his people needs to involve giving the wicked the punishment they deserve. If God were to fail to uphold the demands of retributive justice, then he would be less than morally perfect.

In what follows, I highlight how Craig's line of argument relies on two big assumptions about the nature of value. One of these assumptions is more central to his core argument than the other. But I want to contest both of them. Focusing on the more central assumption Craig makes, I go on to suggest why Christian theists have good reason to reject it.

## 2. Craig's First Assumption about Value

The less central, but still noteworthy, assumption Craig makes has to do with the property of moral *goodness*. He cites John Feinberg's survey of biblical passages that reference God's goodness, offering his own summation: "When the biblical authors speak of God's goodness, what they typically have in mind is not God's moral goodness but God's beneficence or generosity" (Craig 2023, p. 1). It is yet unclear here what "moral goodness" amounts to. But whatever it is, we are to distinguish it from God's beneficence. Regarding God's beneficence, Feinberg himself notes that the biblical descriptions of God's goodness do emphasize the idea that "God is concerned about the well-being of his creatures and does things to promote it". Additionally, though, "God is interested in doing what is

morally good and right". And the biblical writers "capture that idea by referring to his *righteousness* and *holiness*" (Feinberg 2006, p. 366).

Craig shares with Feinberg this working assumption that, when God does something that is "morally good", God is not necessarily acting out of beneficence to further anyone's well-being. But this working assumption is controversial among moral philosophers. A plausible (and, in my view, convincing) line of argument can be offered that all instances of what is *good* are ultimately instances of what is *good for* someone.[1]

Certainly philosophers and theologians writing downstream of Plato have commonly taken *goodness* to be an irreducible, normative, stance-independent property. But a look at how we humans came to appreciate the concept *good* points to a metaphysically less extravagant story. In brief, the story starts with its functional use in Greek society. Alasdair MacIntyre notes that "the word ἀγαθός, ancestor of our *good*, is originally a predicate specifically attached to the role of a Homeric nobleman" (MacIntyre 2003, p. 6). The term was both descriptive and evaluative.

Over time, the conceptual link was lost between the term and the specific description of a Homeric ideal. Changes in societal roles and expectations have naturally led to differing views about which characteristics are ideal or commendable. Accordingly, differing views have emerged as to the conditions under which it is appropriate to offer the positive evaluation that something is "good". And so we arrive at W. D. Ross's conclusion that the best way to understand the historic use of the term is to see it as one of "indefinite commendation" (Ross 2002, p. 66).

Why would we commend anything as "good"? One answer, in the tradition of Plato, is that we recognize it as possessing an irreducible, normative property of *goodness*. Another (and, in my view, more plausible) answer is that we recognize it as benefiting some subject—that is, as being *good for* some subject. Philippa Foot is one who did much this past century to revitalize this more Aristotelian framework that we cannot separate the idea of goodness from the idea of an organism flourishing in its particular form of life (see Foot 2001).

A semantic analysis of the term "good" lends support to the idea that we identify things as good because we see them as good for someone. On Paul Ziff's careful analysis, our use of the term "good" is our indication that something "answers to the interests" of some real or imagined subject (Ziff 1960). Stephen Finlay reaches a similar conclusion by focusing on the structure of our English sentences and the varied ways in which "good" is used in them: *S* is good; *S* is a good *X*; *S* is good for *X*; *S* is good with *X*; *S* is good at *X*-ing; *S* is good for *X*-ing; and so on. Finlay concludes that there *is* a unified logical form to all our uses of the term "good". In keeping with Ziff's analysis, Finlay argues (convincingly, in my view) that our references to things being "good" ultimately make sense only as references to them being "good for" some subject (Finlay 2014).

A semantic analysis of "good" certainly does not establish any firm conclusions about the *nature* of goodness. But it does at least suggest that the property of goodness might be no more than the property of what is good for subjects. I have offered fuller arguments elsewhere for this conclusion about the nature of goodness.[2] In this section, my very brief look at what *goodness* amounts to is not intended to settle long-standing debates. I am only noting that there are controversies to navigate in claiming that God's concern for "morally good" outcomes is not reducible to his concern for subjects' well-being (i.e., for what is good for subjects).

Here is how this point affects Craig's larger line of argument. His discussion of God's righteousness begins by noting that it encompasses two broad and distinguishable concerns: to further the well-being of creatures and to do "what is morally good and right". He goes on to contend that one particular morally good outcome to which a righteous God is sensitive is the outcome of the wicked receiving the punishment they deserve. (I look at this further claim in the next section.) My main point in this section is that Craig's argument—that God's righteousness includes his concern to do what is morally good *in addition to* his beneficent concern for the well-being of creatures—cannot get off the ground if moral goodness simply *is* a matter of what is good for subjects. For there would be no

morally good states to which God could be sensitive beyond those states involving subjects' positive well-being.

### 3. Craig's Second Assumption about Value

The majority of Craig's article is spent on a more particular claim about a specific, non-welfarist value (i.e., a kind of "moral good" not linked to well-being). The value in question is associated with the "punishment of the guilty", which Craig identifies as an "intrinsic good" (Craig 2023, p. 4). This point is my main point of contention with Craig. Actually, the key point of contention is that I reject the idea that divine punishment or discipline has *non-instrumental* value (sometimes called *final* value). And non-instrumental value is, strictly speaking, not quite the same as intrinsic value.[3] But for our purposes this distinction need not concern us.

Could the punishment of guilty persons have non-instrumental value? Craig discusses this question in connection with rival theories of justice. "Retributive theories" affirm that "punishment is justified because the guilty deserve to be punished". In contrast, "consequentialist theories" maintain that "punishment is justified because of the extrinsic goods that may be realized thereby, such as deterrence of crime, sequestration of dangerous persons, and reformation of wrongdoers" (Craig 2023, p. 3).

Now, I am a little uncomfortable with the way Craig frames things here. To speak of "punishment" being "justified" raises a worry in my mind about question begging. "Punishment" can be viewed as, by definition, an exclusively retributive measure—in contrast to the more restorative idea of discipline. I would not want us to assume at the outset that, when God works against the immediate well-being of a wicked person, God is issuing "punishment" in contrast to "discipline". That has to be argued for, not stipulated through terminology. (Craig does not attempt any question begging in this way. I am simply noting that instead of "punishment" I would be more comfortable using a broader term like "hard treatment"—or at least clarifying that "punishment" could be either retributive or restorative.)[4] Also, I would prefer to ask about God's *desired goal* when acting, as opposed to asking about what would "justify" a divine act of punishment. So-called moral justification seems to me a nebulous idea. I think it is much clearer just to ask about the ultimate goals or purposes God is seeking to achieve when acting toward wicked people.

One final quibble before moving on: I would prefer not to equate "prospective" accounts of justice with "consequentialist" accounts of justice, as Craig does. *Consequentialism* as a normative ethical theory is often understood to involve several commitments that I for one would not want to make. The biblical account of justice is, in my view, restorative and not retributive, but I would not want to say that this account commits me to anything like an unvarnished consequentialist theory of ethics.[5] I do think Craig's categories of "prospective" and "retrospective" accounts of justice are helpful, and I am happy to follow that terminology.

These are minor quibbles. Craig is certainly right on the larger, central point that "retrospective" accounts of justice are to be contrasted with "prospective" accounts. My own understanding of God's restorative justice would, as he notes, be prospective. God's (current) punishment or discipline of the wicked would be a means to the (prospective) further end of their repentance and restoration to healthy relationships with God and with others.[6] In contrast, Craig's understanding of God's retributive justice is retrospective.

On to Craig's view. When God exercises (retributive) justice in restoring a "correct moral order in the world" (Craig 2023, p. 2), he looks back at the wicked actions people perform. If God were to explain his motivation in justly punishing the wicked, he would not point to future good outcomes for people. He would point only to people's past wicked actions. Of course, whenever God performs acts directed to the wicked he may *also* be looking forward to future, good outcomes (such as people's restoration). But as regards the motivation to establish *justice* in a right, moral ordering of the world, God would simply be addressing people's past wicked actions (or perhaps their wicked character as displayed in those actions).

So why exactly is God's punishment of the wicked needed as part of God's establishment of a right, moral order in the world? Craig makes the standard move among defenders of retributive justice by insisting that this punishment is *deserved*. God's response to sin must be to impose "justly deserved punishments" (Craig 2023, p. 6). For there is "something of value, something good" in the wicked receiving their deserved punishment (Craig 2023, p. 6). And a morally perfect God is sensitive to this value and acts to realize it.

But *why* is it good, or valuable, that a wicked person receive painful treatment? This is the question that ultimately needs answering. We could of course list various *prospective* reasons: It may cause the person to rethink her self-reliance; it may prompt the person to turn to God; and so on. But such prospective matters, relevant to questions of the possible *instrumental* value of painful treatment, are not at issue. I am asking about the reasons why there may be *non-instrumental* value in a wicked person receiving painful treatment.

We come now to the big assumption that lies behind Craig's central claims about desert and the value of retributive treatment. This assumption is hardly unique to Craig. There is a long history in Western moral philosophy of assuming that there is value in *proportionality itself* obtaining between a person's virtuous/vicious character and the positive/negative treatment she receives.

Sometimes philosophers have been explicit about this value. Kant, e.g., asserted that the "highest good" is happiness "in exact proportion with the morality of the rational beings who are thereby rendered worthy of it" (Kant 1899, p. 456). This past century, W. D. Ross identified "the allocation of pleasure to the virtuous" as one of four "intrinsic goods" (the others being virtue, pleasure, and knowledge) (Ross 2002, p. 140). Ross was at the tail end of what Thomas Hurka calls "a golden age for moral theory", the period in British moral philosophy that began with Sidgwick and included Hastings, Rashdall, Prichard, Broad, and Ewing. Hurka observes that, during this time of rich moral theorizing, "two commonly accepted objective values were virtue and desert", with desert understood, as Ross put it, as "the proportionment of happiness to virtue" (Hurka 2001, p. 6; Ross 2002, p. 27).

Ross recognized that there is no real argument available in support of this (purported) value of proportionality between virtue/viciousness and positive/negative well-being. It is a value, Ross insisted, that we "must recognize" intuitively (Ross 2002, p. 136). And he claimed that we do, in fact, all have a "decided conviction" that it is good when a virtuous person receives happiness in line with his character (Ross 2002, p. 138).

Ross's view represents the received wisdom on desert that has been handed down to us today. Most contemporary philosophers writing on desert affirm that, if a person deserves *X*, then there is intrinsic value in the person receiving *X*.[7] Nathan Hanna refers to this view as "the standard view", observing that "many of those who accept it just assume it" (Hanna 2019, pp. 109–10).

Does Craig assume the standard view? He certainly seems to. We might circle back to Craig's claim that God's moral perfection includes two broad concerns: for the "well-being of his creatures" and for "what is morally good and right". It is difficult to think how this concern for "what is morally good and right" *would not* involve a concern for the (purported) final value of proportionality obtaining between people's character and their treatment. After all, surely no one would claim that there is value in the painful experiences *themselves* of creatures. Rather, as Craig emphasizes, the value at stake is derived from the fact that the suffering of wicked creatures is *deserved*. But since there is not intrinsic value in the suffering itself, the only place for value seemingly has to be in the proportionality that obtains between a creature's suffering and his vicious character.[8] This assumption about value is, again, not a novel one. It is "the standard view" (as Hannah put it) of the value associated with desert. If we do assume the standard view, then Craig's broader line of argument does become quite understandable: God is sensitive to the value of proportionality and performs acts of retributive justice to realize it. Accordingly, God's "moral perfection", best conceived in terms of God's "righteousness", has to comprise "both His *agape* love and His retributive justice" (Craig 2023, p. 6).

What should we make of the standard view? Once again, this view affirms that there is intrinsic value in a person receiving happiness/unhappiness in proportion to her virtuous/vicious character. I do agree with Hannah that this view is the standard one among moral philosophers. I also think Hannah is right that, most times, this view is just assumed. I further think that Christian theists should be the first ones challenging this view!

We Christian theists affirm that God is triune and, thus, essentially relational. We affirm that God's purpose for us is that we participate in relationships with God and with others through which we attain ultimate flourishing. Why think that a relational God would be sensitive to the (impersonal) value of "proportionality itself obtaining" between a person's virtue/viciousness and her positive/negative well-being?

In his defense of the "intrinsic good" realized by God's punishment of the wicked, Craig offers a suggestive moral framework that, to my mind, can be filled out most readily with impersonal moral principles. He notes various biblical passages that reference God's corrective actions and concludes that "The God of the Bible is not just a benevolent father figure but, as Hugo Grotius emphasized in his critique of Faustus Socinus, God is the impartial Ruler and Judge of the creation, responsible for maintaining its moral order" (Craig 2023, p. 4). As to how God's "impartiality" in dealing with the wicked is important in maintaining our world's "moral order",[9] I presume the idea would be something like God establishing proportionality between *every* person's character and his or her respective well-being.[10]

Certainly Western philosophers from the early modern period onward, having moved away from Socrates's framing question of what the good life consists in, have sought to identify various moral principles that might help frame our thinking about how we ought to live.[11] It has been claimed that we should think of justice as grounded in the principles of utilitarianism or contractualism or egalitarianism. Theorists from Kant to Rawls have suggested formulas for discovering, through reason or through intuition, various moral principles that are said really to exist and to be awaiting our discovery. What we have been calling the standard view of desert—that there is value in proportionality itself obtaining between a person's character and her treatment—stands in a line of principles offered by philosophers as grounds for how people ought to be treated. One thing it has in common with these other impersonal principles is that it is not derived from the personal goals God is described in the Bible as having for his creation.

I would argue that scripture depicts God as creating humans for the enduring purpose of having loving fellowship with him (I argued for this conclusion more fully in *But What About God's Wrath?* and I offer a summary defense of it in Section 3). Our participation in the life of God also includes participation in loving, interdependent relationships with other humans and with all the rest of the created order. I find no reason in scripture to think that God would give weight to any moral principle that did not somehow help lead us and the rest of the created order toward these relationships for which we were created. I cannot see why God would be sensitive to the value of any purported moral principle, except where such a principle is an instrumentally valuable guideline in helping bring about those states of final value (which, again, are flourishing relationships as God intended). Accordingly, I cannot see why a Christian theist should want to affirm the standard view of the value of deserved punishment, which I think is pretty clearly assumed within Craig's understanding of a good and right moral order.

Perhaps Craig would want to avoid this conclusion by saying that retributive justice enacted toward the wicked is needed as part of the *vindication* of victims so that these victims might flourish as God intends.[12] In one place, Craig objects to those who "have focused solely on the positive role of God's righteousness in vindicating and saving his people, when in fact the flip side of that vindication is the punishment of the wicked who are oppressing God's people and opposing God" (Craig 2023, p. 2, n. 10). But such a move would be unpromising. The oppressed are not vindicated by the punishment of their oppressors. They are vindicated by the *truth* of their cruel treatment being brought to light and publicly

acknowledged by all parties within a community. Sometimes, proportional treatment does help bring to light an offender's actions or character. ("Wow, that corporation received a huge fine from the EPA. Their environmental abuses must have been really egregious!") At other times, though, proportional treatment fails to bring a truthful narrative to light. And when it does not, we are unsatisfied that proportional treatment has accomplished what we were hoping it would accomplish: namely, the reestablishment of both the victim's and the perpetrator's standings in a community, where these standings accurately reflect who they are and how they have affected one another (see Kinghorn 2021, §6.4–6.6).

Further, there are various ways (other than meting out proportional treatment to oppressors) for God to demonstrate solidarity with the oppressed, to honor them, and to take up their cause. The Book of Revelation is, as I read it, a great *unmasking* of, among other things, coercive powers that masquerade as real or highest powers (which I think Christian theists should take to be God's loving, creative power). If the nature of sin is indeed self-destruction, then oppressors naturally experience destruction when God removes the contingencies that have heretofore allowed them some measure of flourishing (e.g., benefiting from the fruit of oppressed individuals' labor, utilizing the social capital of others' goodwill, etc.). In short, God can vindicate the oppressed, with oppressors experiencing public downfall, without actively punishing oppressors as an act of retribution.[13]

Now, it is true that, if an *oppressor* is to repent at some point and become part of the healthy community God intends for all people, the oppressor's acknowledgement of his own past actions toward others may need to include such elements as apology and, to the extent possible, recompense.[14] This apology and recompense might be costly. Perhaps also there is some role for victims to have a say in whether, and how, the community should respond punitively to the oppressor in restoring what the community understands the shape of healthy relationships to be. Even so, this is again where Christian theists should be the first ones embracing the value of the oppressor's ultimate flourishing. If Christian maturation includes taking up Christ's willingness to extend forgiveness to the repentant, along with God's broader commitment to the goal that all people experience fullness of life eternally, then earthly victims now included within the heavenly community of perfected relationships will surely want their earthly oppressors to repent and find a way into that heavenly community.

So the "vindication of God's people" is an unpromising reason why Christian theists should think it good for oppressors to meet with proportional, negative treatment. Proportional treatment is but one way to bring oppressors' actions to light and to affirm God's perspective on their true histories and on their true standing before him. If the suffering of oppressors is not necessary (or even, at times, sufficient) to vindicate victims, then surely this is not the preferred way for God to finally vindicate all victims (or for victims who have been thoroughly conformed to Christ's likeness to want to be vindicated). From previous discussions, also unpromising (as a reason to think it good for oppressors to suffer) is the assumption behind "the standard view" of desert: that there is intrinsic value in proportionality itself obtaining between a person's virtue/viciousness and her positive/negative well-being. So I conclude that Christian theists have no good reason to affirm—and much reason to reject—any final value in wicked people meeting with proportional suffering.

## 4. Reading the Biblical Witness

Craig is certainly right that many passages in scripture do describe God as responding to persistent human wickedness with actions that both (1) are intended to rectify this sinful warping of God's intentions for humankind and (2) involve painful experiences for those who have acted wickedly. Sometimes, the ultimate motivation behind God's response is clearly described as restorative. ("In vain I punished your people; they did not respond to correction" (Jeremiah 2:30).) Sometimes, there is no mention of any restorative goal. ("But by your hard and impenitent heart you are storing up wrath for yourself on the day of wrath, when God's righteous judgment will be revealed" (Romans 2:5).)[15] The question is

whether God's painful treatment of people is always for some restorative purpose, whether explicitly stated in some particular passage or not.

In *But What About God's Wrath*, I offered both philosophical and biblical reasons for concluding that, when God acts correctively, his ultimate motivation should be viewed as always restorative.[16] Craig, of course, rejects this conclusion, insisting that God sometimes acts retributively and not merely restoratively. In my reply here, I have thus far appealed largely to considerations in value theory in claiming that there is no final value associated with retributive justice to which God might be sensitive. I offer a few comments now on the biblical witness.

I will not attempt examinations of particular passages.[17] Rather, I very briefly just raise some broader, methodological points. First, as J. S. Mill pointed out, matters of "expediency" within a community can readily take on a "character of absoluteness"—and rightly so. Expediency here includes such good ends as the education of children, consensus building within a community about the harm of certain behaviors, and deterrence of those behaviors. To accomplish such expedient aims, our discourse with one another understandably becomes absolutist, deontic, and juridical in nature. Those actions judged especially bad are not merely discouraged but are declared "not to be done, period!" (on threat of punishment). Even if the ultimate motivations for punishment are ones of expediency, a community's language—"If you do this, we will do to you in kind!"—hardly reflects this fact.

Craig is again right in noting that many biblical passages do describe God's response to wickedness with terms that lie within the conceptual category of absolutism and not of expediency. Yet, that point itself surely does not settle the question whether God's ultimate motivation in allowing or causing hard treatment of the wicked is an expedient one (e.g., the expedient end of their repentance and restoration). How do we settle it? Difficult questions of methodology abound. We, of course, need to consider the larger biblical picture of what God has revealed as his plans for humankind and, therefore, what he does and does not value. Perhaps we need to draw from reflections on value theory to help shape our expectations of what God might be valuing in a particular biblical narrative. Perhaps we need to engage thoughtfully with how others in the Christian tradition have interpreted the shape of justice in the Bible.

On this last matter, Craig does identify one trend. Commenting on Jordan Wessling's claim that God's moral attributes are all fundamentally a matter of love, Craig states that this thesis seems to him to be "the lingering vestige of classical liberal theology, which eschewed the justice and wrath of God in favor of love" (Craig 2023, p. 3). He then notes a more recent development: "In contrast to classical liberal theology, neo-liberal theology, if we may coin a term, affirms God's wrath but sees it wholly as a manifestation of His love aimed at the reformation of sinners" (Craig 2023, p. 3).

I, of course, would want to make this last affirmation. But I would not see classical liberal theologians as my natural allies on topics such as God's wrath or God's justice. Instead, I would want to note a different trend in looking for allies. I would want to look at the growth of global Christianity over the past few decades. The center of the tradition has moved southward and eastward. And those of us in North America and Europe have been introduced to an array of theologians and biblical scholars from cultures that do not conceive of justice apart from reconciliation and do not comprehend the human problem to be one of guilt from objective wrongdoing. As a missiology colleague from India recently said to me, these cultures "are relational in how they think about everything!"[18] Understandably, then, many biblical scholars from the global Church just do not find a God in the pages of scripture whose justice is retributive or who, more broadly, bears much resemblance to Grotius's image of an impartial judge.

Another trend I welcome is the theological retrieval movement of the past few decades in which both Roman Catholic and Protestant theologians have focused anew on early Church fathers, especially in the Eastern tradition. Irenaeus, Origen, Athanasius, Gregory Nazianzen, and Gregory of Nyssa are among those early writers who either have no place

for retributive justice or view it as much less central than restorative justice to the kind of moral order God is pursuing (see Reardon 2015). Certainly one can remain tethered to Christian orthodoxy and reject the idea that God is committed to retributive justice as a final, valuable end. I, of course, acknowledge that Craig can make his opposing claims about retributive justice, also well within the boundaries of orthodoxy. But in this debate about retributive justice, I do think a great deal is at stake in terms of how we view the character and commitments of God. In this article, I have drawn from considerations in value theory in order to defend the view that God unceasingly pursues benevolent ends toward us—saints and sinners alike—whenever he acts in pursuit of the outcomes he desires for his creation.

**Conflicts of Interest:** The author declares no conflict of interest.

## Notes

1　David Baggett has suggested to me that there is quite a bit of middle ground between Craig's position and the position that *goodness* is reducible to what is *good for* subjects. One might challenge Craig's views by claiming, more modestly, that whatever is good has as part of its meaning or nature the goodness of someone or other. That proposal would indeed be a challenge to Craig's view, as it would not allow him simply to divorce matters of goodness from matters of goodness for. I pursue here, though, the stronger claim that the property of being *good* is reducible to the property of being *good for* someone.

2　See (Kinghorn 2016, chp. 1 & 2).

3　See (Korsgaard 1983) for the distinction.

4　Regarding the narrower sense of punishment as retribution, there may be a number of conditions required for genuine *punishment* to occur: guilt of the transgressor; the transgressor's recognition that the hard treatment he is receiving is on account of this guilt; the intention of the punisher to give the hard treatment on account of this guilt; the position of the punisher as an appropriate administrator of punishment; and so on (see (Kyle 2013) for a discussion of the conditions for punishment). For present purposes, I ignore this list of conditions and simply assume, with Craig, that there is a sense of "punishment" narrower than "hard treatment" and linked to the (purported) value of retributive justice.

5　Craig notes in passing an objection sometime leveled at "consequentialist theories of justice", namely that "on such theories it may be just to punish the innocent in view of the good consequences" (Craig 2023, p. 5, n. 19). Craig comments that we need "a view of divine justice as retributive, lest God punish the innocent on consequentialist grounds" (Craig 2023, p. 4, n. 17). This has always seemed to me a strange objection—at least if it is used against Christian theists offering restorative theories of justice. If "punishment" simply means hard or painful treatment, then almost all Christian theists actually *do* allow that God, presumably for prospective reasons, does sometimes allow or even cause both the guilty and the innocent to suffer short-term pain, consistent with his longer-term plans to draw them and others into eternal, perfected relationships through which they find their ultimate flourishing. Job is a reminder here. On the other hand, if "punishment" is conceptually tied to "declaring guilty", then in punishing the innocent God would be declaring an untruth. Aside from the dubious idea that God's essential nature is consistent with declaring untruths, surely Christians reject the notion that the God-centered community of perfected relationships—to which God is drawing all people—could be built on untruths about who the people in that community were and are. In short, there seem no prospective reasons why the God of Christian theism would declare the guilty innocent. Thus, I do not see how restorative theories of justice put forth by Christian theists would be at all vulnerable to the kind of objection Craig notes against unnuanced forms of consequentialism.

6　I should note that some of these others may themselves find further restoration in various ways. For example, in correcting an oppressor so that his actions are publicly exposed, God may thereby provide a way for the oppressor's victims to find deeper, healthier relationships with family and friends who now understand them more fully and can stand more fully with them in solidarity. In short, when God disciplines an oppressor, God's aims of restoration need not be limited to the *oppressor's* restoration. Due to limits of space, though, I continue to discuss only the restoration of the wicked as God's goal when he disciplines them.

7　See (Sher 1987, chp. 8 & 11; Pojman 1997, p. 566; Miller 1999, pp. 135–6; Hurka 2001, pp. 7–8; Schmidtz 2006, pp. 85–6; Kristjánsson 2006, chp. 2; McMahan 2009, pp. 8–9; Kershnar 2010, chp. 1 & 6; Kagan 2012, p. 17; Berman 2013, p. 89; Zaibert 2017, pp. 14–16).

8　I suppose one might suggest that it is not *proportionality* here that has value but, rather, *deserved suffering* that has value. Following G. E. Moore's account of "organic wholes", one might claim that the suffering of people with vicious character is an organic whole with positive value—even while the parts that make up that whole (suffering, people with vicious character) each have negative value (see Zaibert 2018, chp. 2). But Moore's intuitive appeal to the supposed goodness of vicious people's suffering only gains traction if one finds a kind of order or appropriateness in this organic whole. And the kind of order in question is surely just proportionality obtaining between vicious character and negative well-being. So we seem back at our question of whether we have reason to think that there is value in proportionality itself obtaining between a person's character and her well-being.

9    As opposed to, e.g., God's "impartiality" in calling both Jew and Gentile to participate in the New Creation, which, incidentally, seems to me to be the clear focus of Paul's emphasis on God's impartiality in Romans—despite Craig's frequent attempts to draw affirmations of God's retributive justice from this Pauline letter.

10   I do not mean to suggest that Craig has no room to say that God, in seeking a right, moral order, desires that the wicked repent and embrace his offer of forgiveness and eternal life in fellowship with him. Craig indeed has room to say that God values and desires the well-being of the wicked. There is even room for Craig to say that God values the restoration of the wicked much *more* than he values the just punishment of the wicked. I am only focused on Craig's commitment to the supposed value of just punishment, wherever it ranks within the list of final values to which God is sensitive. It might also be worth noting that, when God *does* restore the repentant sinner then, on Craig's view, there would seem to be a *trade-off* of valuable states of affairs. Yes, Craig could say that there is enormous—perhaps infinite—value in a sinner's restoration to eternal, abundant life with God. Still, there has been *something* of value lost along the way (recall Craig's statement that there is "something of value, something good" in the wicked receiving their deserved punishment). I myself would much prefer to say that it is *altogether* (intrinsically) bad that sin exists, that suffering exists, and that sinful people persist in suffering—and *altogether* good when these states no longer obtain.

11   See (Darwall 1995). Tellingly, Grotius (whom Craig cited above) is the one whose work, *The Law of War and Peace*, Darwall identifies as "the founding work of modern natural law" (Darwall 1995, p. 5). This modern natural law tradition is to be contrasted with, and was largely a reaction to, the kind of natural law tradition formulated by Aquinas. In this earlier tradition, God's eternal laws "applied and made accessible" to us in the form of natural laws specify "the distinctive perfection or ideal state of every natural thing" (Darwall 1995, p. 5). Framing questions of "what makes for the good life?" and "how ought we to live?" then come to much the same thing. But a moral framework not centered on questions of how to further subjects' well-being goes searching for non-welfarist moral principles to explain why we ought to do certain things. I would also want to raise this historical point in response to a comment Craig makes about my own continued emphasis on God's beneficent goal of our flourishing. Craig cites Laura Garcia in commenting that my emphasis neglects God's justice, which is an aspect of his moral goodness: "A problem here is that, as Garcia notes, flourishing or well-being is a type of non-moral good, and it seems wrong to treat moral value as simply a function of non-moral value (Craig 2023, p. 5, n. 18; cf. Garcia 2008, pp. 221, 229). If by "moral" we mean the modern (downstream of Grotius) quest for a grounding of "the moral ought" in something other than God's telos for us—which is abundant life in communion with God and others—then I suppose my framework for understanding normative concepts (like "good" and "right") does indeed center on non-moral goods. But I would absolutely reject this modern understanding of the nature of normativity with its subsequent framing of the moral/non-moral distinction.

12   Though I note that this rationale for punishing the wicked is different than Craig's earlier rationale that such punishment is an "intrinsic good". For if we claim that punishment achieves the good purpose of restoring the status of victims, then we are claiming that punishment is *instrumentally* valuable.

13   In our current world of imperfectly loving attitudes and limited imagination, it may be that victims often only understand God's message that he is on their side if they see him as acting *against* those who are *not* on their side. But this is a far cry from affirming that God's preferred way of vindicating the oppressed is by punishing their oppressors or that this retributive punishment is somehow conceptually linked with the vindication of victims.

14   See (Swinburne 1989, pp. 81–87) on the conditions for genuine atonement between oppressor and victim.

15   As a quick sidenote, Craig asks why, if I view God's hard treatment of individuals as always restorative, I would not want to claim that God "simply annihilate(s) the damned and put(s) them out of their misery" (Craig 2023, p. 6). In fact, I *am* inclined toward an annihilationalist view of those who have decisively rejected all avenues that God's grace might take toward them. But in my book on wrath, I sought to avoid taking a stance on such eschatological matters as how likely it is that some (or many) people have decisively rejected God; whether postmortem opportunities for repentance may be available; and whether the biblical references to eternal separation from God imply separation from every aspect of the *life* coming from God (i.e., imply death for those truly separated from God). Instead, I sought only to explain the sense in which we can rightly say that those who have decisively rejected God have eternally "placed themselves under God's wrath"—consistent with my overall argument that God's wrath has a restorative purpose.

16   Much more detailed interpretations of the biblical witness as pointing to a restorative view of divine justice, over against a retributive view, include (Hays 1989; Marshall 2001; Travis 2009; Wright 2016).

17   In my book on wrath, I did look at some key passages often interpreted along retributive lines (see also the recent overview of Romans and Isaiah, which very much represents my own position, in (Rutledge 2022, chp. 5)). Beyond what I have said in that book, I think the parable of the Rich Man and Lazarus (Luke 16:19–31) is particularly instructive as to how a passage of scripture that speaks of divine judgment can present us with what looks like a clearly retributive *or* a clearly restorative message—depending on the assumptions we bring to the text as to whether there could be final value in wicked people receiving proportionally negative treatment. I look at this parable in (Kinghorn 2021, pp. 214–15) and, with particular attention to the value of retributive justice, in (Kinghorn forthcoming, §4.5).

18   As was, it is well worth noting, the culture of first-century Palestine and the earlier cultures of the ancient Near East. In sharp contrast to Craig's interpretive framework for understanding various New Testament passages about God's justice, see the more relational and Eastern interpretive lens provided in (W 2019).

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
