# Peer review of "God’s Moral Perfection as His Beneficent Love. Comment on Craig (2023). Is God’s Moral Perfection Reducible to His Love? Religions 14: 140"

_religions, doi:10.3390/rel14091205_

Round 1

Reviewer 1 Report

This is a pretty easy acceptance. Sorry that I do not have any terribly interesting comments or feedback. The paper is just a good, critical reply to Craig. It continues the debate in a constructive way. I think it will be perfect for the special issue. 

Author Response

Thank you for the kind review.

Reviewer 2 Report

"God’s Moral Perfection as His Beneficent Love: Reply to Craig" is clear, charitable, and advances the relevant debate on the nature of divine punishment and love. I recommend publishing it.

However, there is one minor issue that could perhaps be clarified. Within lines 231-236 the following claim is made:

"One thing it [i.e., Craig's view on divine retributive punishment] has in common with these other impersonal principles is that it is not derived from the personal goals God is described in the Bible as having for his creation. Scripture depicts God as creating humans for the enduring purpose of having loving fellowship with him. Our participation in the life of God will also include participation in loving, interdependent relationships with other humans and with all the rest of the created order."

However, whether Scripture does not also attribute the kinds of retributive goals that Craig would want to affirm of God is a controversial matter. My guess is that most biblical scholars would maintain that Scripture does in fact attribute such retributivist goals to God (e.g., indirectly through the Mosaic law and directly through the descriptions of certain past and future divine acts). Regardless, it seems irresponsible for the author simply to make such a claim (i.e., that found in the quote) without doing more to justify it. And the author does have more to say on this score, in section three of the article. So, perhaps the author should flag that this is a complex issue, one that will be returned to in a later section of the article. But I leave this matter to the article's author and the editors at Religions

Author Response

Thank you for the kind review and for the helpful suggestion.